# *SSA4* Mediates Cd Tolerance via Activation of the Cis Element of VHS1 in Yeast and Enhances Cd Tolerance in Chinese Cabbage

**DOI:** 10.3390/ijms252011026

**Published:** 2024-10-14

**Authors:** Han Zheng, Chao Yuan, Tong Bu, Qun Liu, Jingjuan Li, Fengde Wang, Yihui Zhang, Lilong He, Jianwei Gao

**Affiliations:** 1Shandong Key Laboratory of Bulk Open-Field Vegetable Breeding, Ministry of Agriculture and Rural Affairs Key Laboratory of Huang Huai Protected Horticulture Engineering, Institute of Vegetables, Shandong Academy of Agricultural Sciences, Jinan 250100, China; zhenghan1216@163.com (H.Z.); lijj0620@163.com (J.L.); wfengde@163.com (F.W.); zyh_0923@163.com (Y.Z.); 2Key Laboratory of Plant Development and Environment Adaptation Biology, Ministry of Education, School of Life Science, Shandong University, Qingdao 266237, China; yc340434727@163.com; 3College of Life Science, Shandong Normal University, Jinan 250100, China; 18753889619@163.com (T.B.); 19811827686@163.com (Q.L.)

**Keywords:** Cd tolerance, yeast, Chinese cabbage, *SSA4*, POM34, VHS1

## Abstract

Identifying key genes involved in Cadmium (Cd) response pathways in plants and developing low-Cd-accumulating cultivars may be the most effective and eco-friendly strategy to tackle the problem of Cd pollution in crops. In our previous study, *Stressseventy subfamily A 4* (*SSA4*) was identified to be associated with Cd tolerance in yeast. Here, we investigated the mechanism of *SSA4* in regulating Cd tolerance in yeast. ScSSA4 binds to POre Membrane 34 (POM34), a key component of nuclear pore complex (NPC), and translocates from the cytoplasm to the nucleus, where it regulates the expression of its downstream gene, *Viable in a Hal3 Sit4 background 1* (*VHS1*), resulting in reduced Cd accumulation in yeast cells. Additionally, we identified a Chinese cabbage *SSA4* gene, *BrSSA4c*, which could enhance the Cd tolerance in Chinese cabbage. This study offers new insights into the regulatory mechanisms of Cd tolerance in yeast, a model organism, and paves the way for the genetic enhancement of Cd tolerance in Chinese cabbage.

## 1. Introduction

Due to industrial activities and the excessive use of urban wastewater, pesticides, and fertilizers, heavy metal pollution and its associated toxicity to human health have garnered widespread attention [1,2,3]. Cd is considered one of the most highly toxic and representative heavy metals affecting both animals and plants because of its physicochemical characteristics [4,5,6,7,8]. Cd is absorbed by plant roots from soil or water, transferred to above-ground parts, and accumulated in specific organs, posing potential toxicity risks to human health. To ensure food safety, various approaches have been proposed to reduce Cd concentrations, such as soil dressing, chemical remediation, and phytoextraction using crops such as rice and maize [9,10,11,12]. Nevertheless, the development of low-Cd-accumulating cultivars is regarded as the most effective and eco-friendly strategy. Achieving this goal requires a comprehensive understanding of the mechanisms governing Cd absorption and accumulation in plants.

Evidence suggests that Cd disrupts plant cell homeostasis by inducing the accumulation of reactive oxygen species (ROS) and malondialdehyde (MDA), leading to a wide range of physiological, biochemical, and morphological changes [13,14,15]. Cd is known to inhibit enzyme activity by displacing essential metals at enzymatic active sites or binding to sulfhydryl groups in proteins [16,17]. Recent studies have shown that Cd tolerance in plants is linked to a highly complex regulatory network involving numerous genes. Various types of Cd transporters have been identified in plants. For example, ATP-binding cassette (ABC) transporters are reported to pump Cd into vacuoles or out of plasma membranes in yeast and *Arabidopsis* [18,19]. Additionally, heavy-metal associated (HMA) proteins, belonging to the P_1B_-ATPase subfamily, and natural resistance-associated macrophage proteins (Nramps) also play important roles in transporting Cd ions [20,21]. A number of TFs have also been identified as key regulators of signal transduction in Cd stress responses by regulating the expression of genes involved in Cd uptake, transport, and resistance, such as WRKY, myeloblastosis protein (MYB), basic leucine zipper (bZIP), and heat shock transcription factor (HSF) [22,23,24]. In addition, epigenetic regulation via DNA methylation, lncRNAs, miRNAs, and kinases plays an important role in the transcriptional responses to Cd stress [25,26,27,28,29].

Mitochondrial pyruvate carriers (MPCs), key regulators of both substance and energy metabolisms, have been found to form protein complexes composed of AtMPC1 and AtMPC2 (NEGATIVE REGULATOR OF GUARD CELL ABA SIGNALING 1 (AtNRGA1) or AtMPC3), which are responsible for Cd tolerance and reducing Cd accumulation in *Arabidopsis*, as demonstrated in our previous report [30]. In yeast, the mutant strain Δmpc1 exhibits increased sensitivity to Cd stress compared with the wild-type strain [31]. RNA-seq analysis was conducted on both strains, and DEGs were selected for a serial dilution assay after being inserted into a yeast-expressing recombination vector (pGPD::DEGs) for yeast transformation. As a result, *SSA4*, encoding a Heat Shock 70 protein (HSP70), was identified as the downstream gene in the Cd resistance pathway regulated by MPC1.

In this study, we uncovered and elucidated the molecular mechanism of *SSA4* in response to Cd stress in yeast. We further investigated the functions of the *SSA4* gene family in Chinese cabbage under Cd stress. The results reveal that the biological functions of the *SSA4* gene in Chinese cabbage differ significantly from those in yeast, indicating more complex mechanisms. Nonetheless, our findings offer a novel strategy for regulating Cd tolerance in both yeast and Chinese cabbage and lay the groundwork for further development of low-Cd-accumulating vegetable crops.

## 2. Results

### 2.1. ScSSA4 Reduced Cd Accumulation in Yeast

In our previous study, we found that *ScSSA4* can partially recover the Cd sensitivity of *∆mpc1* [31]. It is indicated that the enhanced expression of *ScSSA4* can bolster Cd tolerance in yeast. Furthermore, the yeast cells overexpressing *ScSSA4* accumulated less Cd than the control cells (Figure 1A). Non-invasive micro-test technology was employed to monitor the Cd ion flux across the yeast cell membrane surface (Figure 1B). The result revealed that Cd uptake in yeast cells with the overexpression of *ScSSA4* was reduced compared to the control, thereby explaining the decreased accumulation of Cd in these yeast cells. These results suggested that the Cd-tolerance phenotype mediated by *ScSSA4* may primarily be due to the decreased Cd accumulation.

### 2.2. ScSSA4 Enters the Nucleus under Cd Stress

To investigate the specific role of *ScSSA4* in Cd stress tolerance, the CDS was cloned into vector pRS416-GFP and expressed in yeast cells, and EV (pRS416-GFP empty vector) was taken as the negative control. ScSSA4 was mainly localized in the cytoplasm in the absence of Cd. However, under Cd stress, ScSSA4 translocated into the nucleus while the negative control was not affected by Cd (Figure 2). The results suggest that ScSSA4 could interact with the *cis* elements of its downstream target genes located in the nucleus and is required for the yeast Cd stress response.

### 2.3. ScSSA4 Interacts with ScPOM34

To explore the function of ScSSA4 in yeast, a yeast two-hybrid (Y2H) screening experiment was conducted for candidate interacting proteins. The cDNA library of wild-type yeast was constructed for this study. The full length of *ScSSA4* was fused into pGBKT7 as a bait for screening. Several independent positive proteins were isolated, and ScPOM34 was identified as an interacting protein. AD-ScPOM34 and BD-ScSSA4 were co-transformed into Y2H Gold yeast, and the yeast dilution assay was tested on an SD/-Leu/-Trp/-His/-Ade plate. These results indicated that there is an interaction between ScSSA4 and ScPOM34 (Figure 3A). To confirm the interaction between ScSSA4 and ScPOM34, co-immunoprecipitation (Co-IP, Figure 3B) and bimolecular fluorescence complementation (BiFC, Figure 3C) assays were performed. These results also revealed that ScSSA4 could interact with ScPOM34. 

### 2.4. Screening Candidate Genes Involved in the Pathways Mediated by SSA4 through Transcriptom Analysis

Considering *ScSSA4* can significantly recover Cd sensitivity of *∆mpc1*, but does not further improve Cd tolerance in the wild-type strain (W303a), RNA-Seq analysis was performed on W303a-EV, W303a-SSA4, *∆mpc1*-EV, and *∆mpc1*-SSA4 to identify specific downstream genes involved in the *SSA4*-mediated regulatory pathway. DEGs were selected according to the following criteria: log_2_ (fold change) > 1 or <−1/2 and padj ≤ 0.05. By comparing the gene expression levels in both the W303a group (W303a-EV VS W303a-SSA4) and the *∆mpc1* group (*∆mpc1*-EV vs. *∆mpc1*-SSA4), 101 and 147 DEGs were identified, respectively. Of these, 128 DEGs were common to both groups, including 9 upregulated and 119 downregulated genes (Figure 4A,B). We hypothesize that the nine upregulated genes may play specific roles in the *SSA4*-mediated regulatory pathway, with expression patterns likely similar to those of *SSA4*.

These nine up-regulated genes, named DEG1 to DEG9, were selected for the qRT-PCR analysis to validate the authenticity of the RNA-seq data. These genes include gene-CENPK1137D_680, gene-CENPK1137D_1620, gene-CENPK1137D_1667, gene-CENPK1137D_2938, gene-CENPK1137D_2995, gene-CENPK1137D_4071, gene-CENPK1137D_4082, gene-CENPK1137D_4516, and gene-CENPK1137D_4598. The relative expression levels measured by qRT-PCR are shown in Figure 4C. The results indicate that the DEGs exhibited similar expression patterns between RNA-seq and qRT-PCR, providing a foundation for further functional studies on these nine genes. 

We found that there were nine pathways significantly enriched by DEGs in the group of *∆mpc1*-EV vs. *∆mpc1*-SSA4 (Appendix A). According to the number of DEGs, the top three pathways are “Glycerophospholipid metabolism”, “Glycerolipid metabolism”, and “2-Oxocarboxylic metabolism”. However, it is interesting that there was no significantly enriched KEGG pathway identified in the group of W303a-EV VS W303a-SSA4 (Appendix A). The results indicate that the Cd stress regulation pathway mediated by *SSA4* could be dependent on *MPC1*, and finally, we may have uncovered a novel Cd tolerance mechanism through which yeast responds, and yet further investigation should be required.

### 2.5. Overexpression of VHS1 (DEG6) Leads to Cd Sensitive in Yeast 

In order to investigate the functions of the candidate genes in Cd response, the CDSs of DEGs were inserted into the pRS416-GFP vector and transferred into yeast cells. 

Only the yeast cells overexpressing DEG6, which encodes a serine kinase protein named VHS1, exhibited higher sensitivity to Cd stress compared with the wild-type control (Figure 5A). Moreover, yeast cells overexpressing *VHS1* accumulated more Cd than the control (Figure 5B), suggesting that the Cd sensitivity induced by *VHS1* may be primarily due to increased Cd accumulation. A ChIP-qPCR assay was conducted to investigate the interaction between ScSSA4 and the promoter of *VHS1* in yeast cells that carry the *ScSSA4*-expressing pRS416-GFP vector. DNA was immunoprecipitated using a GFP antibody, and five primer pairs were specifically designed for qPCR to target different regions of the *VHS1* promoter (Figure 5C). The results show that there is a positive interaction between ScSSA4 and the F5 fragment of the *VHS1* promoter. These fundings suggest that ScSSA4 specifically binds to a distinct region of the *ScVHS1* promoter to modulate its expression under the treatment of Cd. 

### 2.6. Expression Pattern of the Chinese Cabbage SSA4 Family Members against Cd Stress

BlastP analysis was conducted using NCBI with the ScSSA protein sequence. Evolutionary analysis was performed with the retrieved protein sequences to eliminate duplicates. Ultimately, thirteen members of the *SSA4* family were identified in Chinese cabbage (Figure 6A) and assigned according to their distribution in the genome (Appendix A). 

Exon/intron regions of the *BrSSA4s* were identified by aligning the CDSs to the genomic sequence. The results show that all *BrSSA4* sequences contained introns except *BrSSA4f*, *BrSSA4h*, *BrSSA4i*, and *BrSSA4m*, and the number of introns ranges from zero to thirteen. Additionally, the number of introns in the same subfamily of *BrSSA4s* was not always the same. For example, *BrSSA4c* and *BrSSA4l*, which belong to the same subfamily, contained five and four introns, respectively (Figure 6A). 

The expression patterns of *BrSSA4s* were analyzed at various time points under Cd stress. The results revealed that only five members of the family responded to the Cd treatment, with their expression levels peaking at 3 h post-treatment. Except for *BrSSA4e*, which can maintain a high expression level for an extended period, the expression levels of *BrSSA4b*, *BrSSA4c*, *BrSSA4h,* and *BrSSA4i* declined rapidly after reaching the peak. These results suggest that the five members could be involved in the molecular regulatory processes of Chinese cabbage in response to Cd.

### 2.7. BrSSA4c Is Involved in the Cd Stress Response of Chinese Cabbage 

To verify the functions of these five *SSA4* genes in Chinese cabbage, transgenic lines overexpressing each gene were generated and identified (Figure 7C–E). Surprisingly, only the transgenic Chinese cabbage overexpressing *BrSSA4c* exhibited a significant phenotype under Cd stress (Figure 7). Interestingly, in the absence of Cd, transgenic Chinese cabbage overexpressing *BrSSA4c* displayed noticeably shorter root length compared to that of the WT. Under Cd exposure, the inhibition of transgenic plants was significantly weaker than that of WT plants. These results suggest that *BrSSA4c* could be an important regulatory element enhancing Cd tolerance in Chinese cabbage.

## 3. Discussion

Cd is widely recognized as one of the most toxic contaminants, prevalent in certain fields where it can cause significant damage to plants. The accumulation of Cd in crops can significantly reduce yields and ultimately pose a threat to human health through the food chain. Understanding the impact of Cd stress on crops and the mechanisms of Cd tolerance holds substantial theoretical value. Although various genes have been identified as participants in Cd absorbance, transport, and accumulation in plants, the critical genes modulating Cd stress remain limited.

Yeast mutants have been widely used to identify the functions of genes under heavy metal stresses. Our previous study shows that overexpression of the transcription cofactor *SSA4* can enhance the Cd tolerance of yeast cells [31]. Measurements of Cd content in yeast cells and Cd ion flux on cell surfaces indicated that *ScSSA4* likely enhanced Cd tolerance by reducing Cd accumulation in yeast cells (Figure 1). Many studies have indicated that Cd tolerance arises from the reduced Cd accumulation, which is attributed to differential gene expression. For example, heterologous expression of *TaWRKY70* and *ZmWRKY64* in *Arabidopsis thaliana* leads to the down-regulation of a number of Cd uptake and transporter genes, resulting in low Cd accumulation in the leaves and conferring Cd stress tolerance in transgenic plants [32,33].

Transcription factors (TFs) are key molecular regulators that control the expression of downstream genes in organisms and can translocate to the nucleus in response to specific signals or stresses. The AtbZIP24-GFP fusion protein is localized in both the nucleus and cytoplasm of plants under non-stress conditions but preferentially targets the nucleus under salt stress [34]. RING DOMAIN LIGASE2 (RGLG2), a RING E3 ligase, translocates from the plasma membrane to the nucleus and interacts with ETHYLENE RESPONSE FACTOR 53 (ERF53) under drought stress in *Arabidopsis* [35]. Similarly, ScSSA4, which functions similarly to TFs, translocates from the cytoplasm to the nucleus under Cd stress (Figure 2). 

Through yeast library screening, we identified a candidate protein, ScPOM34, that interacts with ScSSA4, and validated their interaction using BiFC and Co-IP assays (Figure 3B,C). ScPOM34, localized exclusively in NPC, is an important component of the inner ring of NPCs (IRC) [36]. NPC is a protein complex that creates channels through the nuclear membrane, facilitating communication between the nucleus and cytoplasm. The NPC is composed of multiple copies of more than 30 nucleoporin proteins (NUPs) that penetrate the nuclear envelope [37]. The IRC, as a structural anchor for other NPC components, includes various NUPs, such as NUP205, NUP188, NUP155, NUP135, NUP93, and NUP88. In plants, mutation of NUP205 results in defects in both circadian rhythm and immune response, alongside altered nuclear accumulation of mRNAs related to circadian rhythm and immune defense [38]. Loss of NUP88 function may facilitate the nuclear export of MITOGEN-ACTIVATED PROTEIN KINASE 3 (MPK3), reducing its nuclear accumulation and ultimately compromising fungal resistance [39]. However, the role of POM34 in plant growth, development, and stress responses remains largely unreported.

It is believed that targeting genes to nuclear speckles, lamina, or inner-chromosomal clusters requires cis-acting DNA elements, or TFs [40,41,42]. Similarly, targeting of genes to the yeast NPC upon activation depends on TFs that bind to their promoters [43]. Five yeast TFs, including Proline UTilization 3 (Put3), Suppressor gene for FLocculation (Sfl1), General Control Nonderepressible 4 (Gcn4), STErile 12 (Ste12), and Centromere Binding Factor 1 (Cbf1), from distinct families and with distinct mechanisms of regulation, have been identified to play an important role in promoting the interaction with the NPCs and inducing inter-chromosomal clustering [44]. Several inducible genes are found to be highly expressed under specific environmental conditions, such as *GALACTOSE KINASE 1* (*GAL1*), *HEXOKINASE 1* (*HXK1*), *INOSITOL 1-PHOSPHATE SYNTHASE* (*INO1*), *HSP104*, and *TRYPTOPHAN SYNTHASE ALPHA CHAIN 2* (*TSA2*). Proteins encoded by the genes aforementioned are localized within the nucleus in the absence of stress but are recruited to the nuclear pore upon the treatment [45,46,47,48,49]. In animal cells, Ser62 phosphorylation and Peptidyl-prolyl cis/trans isomerasr 1 (PIN1)-mediated regulation control the spatial distribution of MYC in the nucleus, where MYC has been shown to interact with Nup153 (nuclear pore basket) [50]. Currently, there are no reports on the interaction between SSA4 and any components of NPCs, and its function remains to be further understood.

Consequently, the transcriptome was utilized for screening the downstream target genes of *ScSSA4*, and a candidate gene named *VHS1* was identified (Figure 4). ChIP-qRCR also confirmed the interaction between the promoter of *VHS1* and SSA4 (Figure 5). *VHS1* encodes a cytoplasmic serine/threonine protein kinase. Sucrose NonFermenting 1 (SNF1) in yeast, an AMP-activated protein kinase, is a crucial regulator of metabolism, which is activated when energy is exhausted. The activity of SNF1 in the nucleus is regulated by VHS1 and its substrate SOS Interaction Protein 5 (Sip5) [51]. It has also been reported that high expression of several genes, such as *Phosphatase type Two C1* (*PTC1*), *CycLiN 3* (*CLN3*), *Na^+^/H^+^ Antiporter 1* (*NHA1*), *Protein PHosphatase3* (*PPH3*), and *VHS1,* improved the growth of yeast cells at high temperatures, and the cells lacking VHS1 were sensitive to high concentrations of NaCl [52]. In this study, we found that *VHS1* was associated with Cd stress. However, the mechanism needs to be further explored. 

It is well known that both *Brassica juncea* (with the AABB genome) and *Brassica napus* (with the AACC genome) are susceptible to heavy metal contamination. *B. napus* (with the AACC genome) is formed through the hybridization of *B. rapa* (with the AA genome) and *B. oleracea* (with the CC genome), while *B. juncea* (with the AABB genome) is formed through the hybridization of *B. rapa* and *B. nigra* (with the BB genome) [53]. The characteristics related to heavy metal content of *B. juncea* and *B. napus* may be attributed to their shared *B. rapa* genes. Given that *SSA4* can reduce the Cd content in yeast cells, we attempted to investigate its effect on Cd accumulation in Chinese cabbage. Chinese cabbage (*Brassica rapa* subsp. *pekinensis*) is a crucial leafy vegetable in East Asia. The leaves exhibit a high capacity for water absorption, along with Cd accumulation, in comparison with other leafy vegetables [54,55]. Therefore, it is crucial to identify candidate genes related to Cd response and their regulatory mechanisms in Chinese cabbage. There were 13 members in the *SSA4* family in Chinese cabbage, among which 5 showed significant responses to Cd treatment (Figure 6). These 5 members were transferred into the wild type of Chinese cabbage, and only *BrSSA4c* showed a Cd-resistance phenotype (Figure 7). Phylogenetic trees and motif analyses have shown that Mitogen-Activated Protein Kinase Kinases (MKKs) and Mitogen-Activated Protein Kinases (MAPKs) are evolutionarily conserved in *Brassica juncea* (*Bju*), *Brassica napus* (*Bna*), *Brassica nigra* (*Bni*), *Brassica oleracea* (*Bol*), and *Brassica rapa* (*Bra*). However, their functions diverged during growth and development [56]. Among the *Calcium dependent protein kinases* (*CPKs*) in these species, including 51 *BraCPK*, 56 *BniCPK*, 56 *BolCPK*, 88 *BjuCPK*, and 107 *BnaCPK* genes, 4 *BraCPKs*, 14 *BolCPKs*, and 31 *BnaCPKs* were identified to be involved in the defense response in *Plasmodiophora brassicae* (*P. brassicae*), indicating that *CPK* genes evolved independently with functional divergence in *Brassica* species [57]. Thus, we speculate that during the evolutionary process, the BrSSA4 family members may be involved in various aspects of plant growth and development, as well as in response to adverse environmental stresses. 

## 4. Materials and Methods

### 4.1. Plant Growth and Treatments

“Guangdongzao” a purebred line of Chinese cabbage, is a preserved variety retained by the Vegetable Research Institute of Shandong Academy of Agricultural Sciences, which exhibits distinct characteristics, including susceptibility to bolting, a shortened growth cycle, and reduced root elongation under Cd stress.

Seeds of “Guangdongzao” were treated with 8% sodium hypochlorite (Sinopharm, Shenyang, China) for 5 min at room temperature and then washed with sterile water five times. These seeds were then germinated in half-strength MS media with 50 μM CdCl_2_ in a greenhouse with a day/night cycle of 16 h/8 h at 20 ± 2 °C. The phenotypes were observed and photographed after 14 days.

For expression pattern analysis of *BrSSA4s*, the seeds of “Guangdongzao” were placed in a petri dish with clean water at 20 ± 2 °C for germination. After that, seedlings were transferred to pots containing growth medium (the ratio of nutrient soil to vermiculite is 1:2 without fertilization) and grown in the greenhouse. Three-week-old seedlings exhibiting similar size were selected for treatment. The experimental group was irrigated with 50 μM CdCl_2_ (Aladdin, Shanghai, China)_,_ while the CK group was watered with distilled water until the growth medium was saturated with the solution. The leaves were collected after 0, 3, 6, 12, and 24 h of the treatment and instantly frozen in liquid nitrogen for RNA isolation. The experiment was performed using three biological replicates.

### 4.2. Total RNA Extraction and qRT-PCR Expression Analysis

Total RNA was extracted from Chinese cabbage leaves using TRNzol Universal (TianGen, Beijing, China) and treated with RNase-free DNase Ⅰ (Takara, Dalian, China) for 45 min. The M5-EASTspin Yeast RNA Rapid Kit (Mei5Bio, Beijing, China) was used for yeast RNA extraction.

First-strand cDNA was synthesized using a FastKing gDNA Dispelling RT SuperMix (TianGen, Beijing, China). qRT-PCR was performed using a Real-time PCR mix (SYBRgreen) (Mei5Bio, Beijing, China) on a Real-time PCR detection system (Roche, Basel, Switzerland). The total reaction volume was 20 µL, containing 10 µL 2×SYBRgreen mix, 1 µL cDNA, 0.5 µL of each forward and reverse primers, and 8 µL distilled water. The cycling conditions of qRT-PCR comprise an initial polymerase activation step at 95 °C for 1 min, followed by 40 cycles, each consisting of 95 °C for 10 s and 60 °C for 30 s. Finally, the dissociation curve was designed to assess the specificity of the product. Each biological replicate sample is tested three times to calculate the average Ct value. The comparative 2^−ΔΔCt^ method was used to calculate the relative expression levels of genes. Primers used in qRT-PCR are listed in Appendix A. *BrACT1* has been successfully used as an internal control in the study by Lee et al., in 2013 [58]. We identified the CDS sequences of yeast genes and the control gene (*Actin*) in the S288C reference genome on NCBI and an online tool named GenScript was used to design qRT-PCR primers. Available online: https://www.genscript.com/tools/real-time-pcr-taqman-primer-design-tool (accessed on 11 January 2022).

### 4.3. Gene Clone and Plasmid Construction

The CDS of *BrSSA4s*, cloned from the cDNA of Chinese cabbage “Guangdongzao”, was inserted into the vector pCAMBIA3300-GFP, respectively.

For the yeast dilution assay, CDSs, such as *ScSSA4*, *ScVHS1*, and *ScHSP78*, were cloned into vector pRS416-GFP to form overexpression vectors.

For yeast two-hybrid experiments, CDS of *ScSSA4* and *ScPOM34* was inserted into the vectors pGBKT7 and pGADT7, respectively. For Co-IP, CDS of *ScSSA4* and *ScPOM34* was inserted into the vectors pCAMBIA1307-MYC and pCAMBIA1307-Flag, respectively. For BiFC, CDS of *ScSSA4* and *ScPOM34* was inserted into the vector pDOE01 at BamH1 and SnaB1 sites, respectively. 

Primers for gene clone and vector conduction are listed in Appendix A.

### 4.4. Yeast Strains Construction

Two yeast strains, the wild type W303a and *∆mpc1* (MPC1 deletion mutant), were used in this work. For transcriptome analysis, pRS416-GFP-Empty vector (pRS416-GFP-EV) and pRS416-GFP-ScSSA4 were transformed into both yeast strains using the LiAc transformation method (Takara, Dalian, China). The positive clones were selected on SD-Leu medium (synthetic dextrose medium without leucine) (Coolaber, Beijing, China) and designated as W303a-EV, W303a-SSA4, *∆mpc1*-EV, and *∆mpc1*-SSA4. Primers for vector conduction are listed in Appendix A.

### 4.5. Transcriptome Sequencing of Yeast

The cells of four lines of yeast named W303a-EV, W303a-SSA4, *∆mpc1*-EV, and *∆mpc1*-SSA4 were harvested for transcriptome sequencing. Three independent biological replicates of each sample were collected. The RNA Nano 6000 Assay Kit of the Bioanalyzer 2100 system (Agilent Technologies, CA, USA) was used for RNA integrity control. The purification of mRNA from the total RNA was conducted using poly-T oligo-attached magnetic beads. In First Strand Synthesis Reaction Buffer, fragmentation of mRNA was conducted using divalent cations under elevated temperature. Random hexamer primer and M-MuLV reverse transcriptase (RNase H-) (NEB, Ipswich, MA, USA) were used for the synthesis of the first strand cDNA. Subsequently, DNA polymerase I (Thermo Fisher Scientific, Carlsbad, CA, USA) and RNase H (Thermo Fisher Scientific, Carlsbad, CA, USA) were used for the second strand cDNA synthesis. Remaining overhangs were converted into blunt ends by exonuclease or polymerase. The DNA fragments with adenylated 3′ ends are linked to the adapter with a hairpin loop structure for hybridization. AMPure XP system (Beckman Coulter, Beverly, MA, USA) purifies the library fragments and obtains fragments of preferentially 370~420 bp in length. Then, PCR was carried out using Phusion High-Fidelity DNA polymerase (Thermo Fisher Scientific, Carlsbad, CA, USA), Universal PCR primers, and Index (X) Primers. At last, the PCR products were purified by the AMPure XP system and assessed on the Agilent Bioanalyzer 2100 system in order to confirm the library quality. The index-coded samples clustered on a cBot Cluster Generation System (Illumina, San Diego, CA, USA) using TruSeq PE Cluster Kit v3-cBot-HS (Illumina, San Diego, CA, USA). Then, the library preparations were sequenced on the Illumina Novaseq platform (Illumina, San Diego, CA, USA) for the generation of 150 bp paired-end reads. Clean reads were obtained by removing reads containing adapter or ploy-N and reads with low quality from raw data using in-house Perl scripts. Hisat2 (v2.0.5) was used for the alignment between the clean reads and the reference genome. The mapping reads were assembled by StringTie (v1.3.3b) [59] using a reference-based manner. FeatureCounts (v1.5.0-p3) counted the number of reads mapped to the same gene, based on which the FPKM for each gene were calculated. Differential expression analysis between two groups was carried out by the DESeq2 R package (1.20.0). In order to control the false discovery rate, the Benjamini and Hochberg’s method was used to adjust the *p*-values. Genes with an adjusted *p*-value ≤ 0.05 were considered DEGs. KEGG analysis was performed using the Cluster Profiler R package.

### 4.6. Subcellular Localization Assay

Yeast cells carrying pRS416-GFP-expressing vectors were cultured overnight in SD-Ura fluid medium (Coolaber, Beijing, China) under an agitation rate of 250 rpm at 30 °C with yeast cells transformed with an empty pRS416-GFP vector as a control. The solutions were diluted with SD-Ura fluid medium to an OD_600_ of 0.1 and cultivated for 6–8 h. Then the solutions were treated with or without 50 μM of CdCl_2_ at an OD_600_ of 0.3. The yeast cells were observed under a Leica DMi8 confocal microscope (Leica, Wetzlar, Germany) after 2 hours’ cultivation. 

### 4.7. ChIP and ChIP-qPCR Assay

Yeast cells with pRS416-GFP-ScSSA4/pRS416-GFP-EV were cross-linked by 1% formaldehyde for 20 min at 30 °C, and the reaction was stopped with 125 mM glycine (Sigma, St. Louis, MI, USA) for 5 min. Yeast cells were collected after centrifugation and washed twice with PBS buffer (11.9 mM phosphate buffer (Coolaber, Beijing, China), 2.7 mM potassium chloride (Aladdin, Shanghai, China), and 137 mM sodium chloride, pH 7.4 (Aladdin, Shanghai, China). The yeast cells were frozen immediately in liquid nitrogen and stored at −80 °C. Nucleic acid extraction by enzymatic lysis was carried out according to the Fission Yeast Handbook. Chromatin was sheared by sonication to an average size of 500 bp using a M220 sonicator (Covaris, Woburn, MA, USA). The immunoprecipitation and DNA recovery procedures were performed as described previously [60]. The immunoprecipitated DNA fragments were quantified by qPCR. The detailed description of the qPCR method was presented in Section 4.2. Primers for ChIP-qPCR were listed in Appendix A. All experiments were repeated three times.

### 4.8. Y2H Screening and Assays

The yeast cDNA library was constructed by Nanjing Ruiyuan Biotechnology Co., Ltd. (Nanjing, China). The PGBKT7 vector with the CDS of *ScSSA4* inserted was used as the bait. Y2H screening was performed on the cDNA library using a yeast transformation system (Clontech, Takara, Japan). ScPOM34 was identified as the interacting protein, and its CDS was cloned into the pGADT7 vector for the validation. The primers used in this investigation are listed in Appendix A.

### 4.9. Agrobacterium-Mediated Transformation

*Agrobacterium* strain GV3101 was used to infect the plants. Single clones of *Agrobacterium tumefaciens* GV3101 carrying different vectors were cultured in YEP liquid medium containing 10 μg/mL rifampicin (Sigma, St. Louis, MO, USA) and 50 μg/mL kanamycin (Sigma, St. Louis, MO, USA) for more than 24 h at a speed of 200 rpm at 28 °C to achieve a high density. A total of 200 μL cells were added to 6 mL YEP liquid medium containing 20 μM acetosyringone (Sigma, St. Louis, MO, USA), 10 μg/mL rifampicin, 10 mM MES buffer (Sigma, St. Louis, MO, USA), and 50 μg/mL kanamycin, and grew for another 8 h. The cells were collected by centrifugation, re-suspended in infiltration medium to reach an OD_600_ of 0.8, and then incubated at room temperature for 3 h. Infiltration was carried out with healthy leaves using a needless syringe. The plants were kept in the dark overnight and returned to the growth chamber the following day. 

### 4.10. BiFC Assay

The *Agrobacterium* strain GV3101 carrying the recombinant plasmid pDOE01-ScSSA4-ScPOM34 was transformed into *N. benthamiana* [61]. Fluorescence imaging was conducted using a confocal laser scanning microscope (Leica DMi8, Wetzlar, Germany) at 45 h after agroinfiltration. 

### 4.11. Co-IP Assay

The *Agrobacterium* strain GV3101 carring the pCAMBIA1307-Myc-ScSSA4 and pCAMBIA1307-Flag-ScPOM34 were co-transformed into *N. benthamiana*. 

The leaves were grinded into powder in liquid nitrogen and homogenized in IP buffer (final concentration: 150 mM NaCl (Sigma, St. Louis, MO, USA), 5 mM DTT (Thermo Fisher Scientific, Carlsbad, CA, USA), 50 mM Tris-HCl, pH 7.5 (Thermo Fisher Scientific, Carlsbad, CA, USA), 1% Triton X-100 (Sigma, St. Louis, MO, USA), and 1 protease inhibitor tablet (Roche, Vacaville, CA, USA)for 10 mL IP buffer). Extractions were clarified by centrifugation at 12,000 rpm for 30 min at 4 °C. A total of 500 µL of protein extract was incubated with 30 µL ProtainA/G magnetic beads and 3 µL anti-MYC antibody (ABCAM, Cambridge, UK) for 16 h at 4 °C in a top-to-end rotator, using another 500 µL of protein extract without any antibodies as a control. The beads were washed four times with pre-cooled IP buffer and then eluted by boiling in protein loading buffer. Immunoblot analysis was carried out on the eluted proteins with an anti-FLAG antibody.

### 4.12. Non-Invasive Micro-Test Technology (NMT)

A glass slide with adhered yeast cells was placed under a microscope, and a Cd²⁺ flow sensor was positioned approximately 10 μm above the cell surface. Each sample is tested for 5 min, with 6 replicates per group. The imFluxes V2.0 software (YoungerUSA LLC, Amherst, MA 01002, USA) was utilized for reading Cd^2+^ flow rate data, which was expressed in picomoles per square centimeter per second. (pmol cm^−2^ s^−1^). Positive values indicate efflux, while negative values indicate absorption. The analysis was conducted by Xuyue (Beijing) Sci. & Tech. Co., Ltd., Beijing, China. 

### 4.13. Western Blot

Protein samples were loaded onto 12.5% SDS-PAGE gels for electrophoresis in order to separate the proteins. Then the proteins were transferred from the SDS-PAGE gels to Amersham HybondTM-P (GE Healthcare, New York, NY, USA) using a Mini-PROTEAN Tetra system (Bio-Rad). The membranes were washed with TBST buffer (final concentration: 20 mM Tris-Cl (Thermo Fisher Scientific, Carlsbad, CA, USA), 150 mM NaCl (Sigma, St. Louis, MO, USA), and 0.05% Tween 20 (Sigma, St. Louis, MO, USA) and then blocked with TBST containing 5% (W/V) nonfat milk (TBSTM) at room temperature. The membranes were incubated with anti-GFP (1:5000) (ABCAM, Cambridge, UK) in fresh TBSTM overnight at 4 °C. After that, the membranes were washed 4 times (5 min each) and then incubated with goat anti-mouse IgG (ABCAM, Cambridge, UK) in newly prepared TBSTM for about 3 h at room temperature. The protein bands were monitored with the ECL substrate (Thermo Fisher Scientific, Carlsbad, CA, USA) after washing for 6 times with TBST.

### 4.14. Statistical Analysis

SPSS version 20.0 (SPSS, Inc., Chicago, IL, USA) was used for statistical analysis. All data were presented as the mean ± standard deviation. The significance of differences between the experimental groups and controls was tested using Paired samples *t*-test. *p* < 0.05 was considered statistically significant with “*” means *p* < 0.05, “**” means *p* < 0.01, “***” means *p* < 0.001.

## 5. Conclusions

Upon Cd stress, ScSSA4 translocated from the cytoplasm into the nucleus, where it bound to POM34, an important component of NPCs. ScSSA4 positively regulated yeast Cd tolerance by reducing Cd accumulation in yeast cells through activating its downstream target gene, *ScVHS1*, which is a cytoplasmic serine/threonine protein kinase (Figure 8). The regulatory mechanism of Cd tolerance in the model organism yeast was further investigated. *BrSSA4c*, a homologous gene of *ScSSA4* against Cd stress, was identified in Chinese cabbage. Although the mechanism remains to be uncovered, our findings in this study provide a promising strategy for stably increasing Cd tolerance in Chinese cabbage and possibly other crops by altering *SSA4* gene expression using a gain-of-function approach.

## Figures and Tables

**Figure 1 ijms-25-11026-f001:**
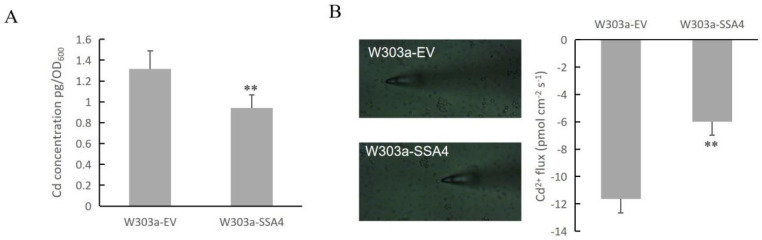
Cd concentrations in yeast cells. (**A**) The two yeast strains were grown on SC solid plates with 50 µM of CdCl_2_ at 30 °C for 5 days. Cells were collected, and the OD_600_ was recorded before the atomic absorption spectrometer measurements. (**B**) Non-invasive micro-test technology was carried out to test Cd ion flux across the membrane surface of yeast cells. Yeast cells overexpressing *ScSSA4* and *ScVHS1* were used in this experiment, and strains with the empty vector pRS416-GFP were used as the control. Six replicates were observed for each strain. Error bars indicate the ± SD of the three independent experiments for each sample. The significance of the difference was evaluated using a paired samples *t*-test by SPSS version 20.0 (** means *p* < 0.01).

**Figure 2 ijms-25-11026-f002:**
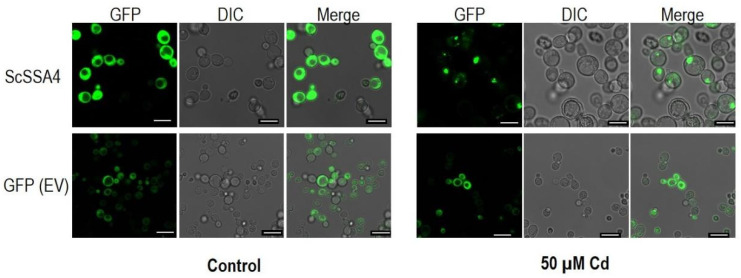
Subcellular localization of GFP-tagged ScSSA4. Yeast cells were treated with or without 50 µM of CdCl_2_. GFP and DIC channels were used to capture the images. Scale bar: 10 μm.

**Figure 3 ijms-25-11026-f003:**
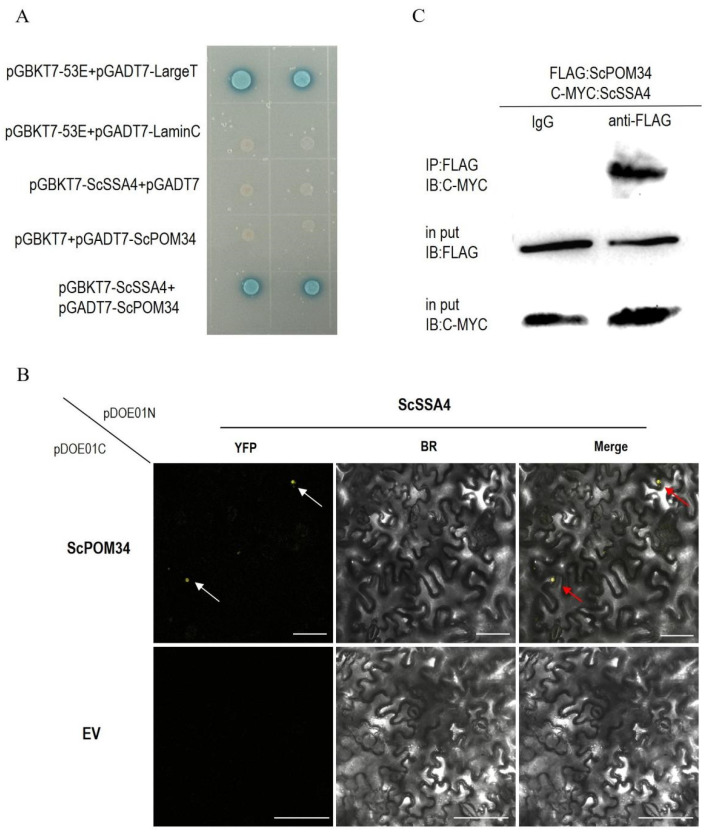
Interaction between ScSSA4 and ScPOM34. (**A**) Y2H assays showing the interaction between ScSSA4 and ScPOM34. (**B**) BiFC assay confirming the interaction between ScSSA4 and ScPOM34. YFP field indicates fluorescence signals. Scale bar: 75 µm. The white arrows indicate the fluorescence signal in the YFP channel, while the red arrows denote the fluorescence signal in the merged image. (**C**) Co-IP assay verifying the interaction between ScSSA4 and ScPOM34. The fusion proteins, ScSSA4-MYC and ScPOM34-FLAG, were co-expressed in tobacco leaves. Immunoblot analysis confirmed the expression of ScSSA4-MYC and ScPOM34-FLAG in the input proteins.

**Figure 4 ijms-25-11026-f004:**
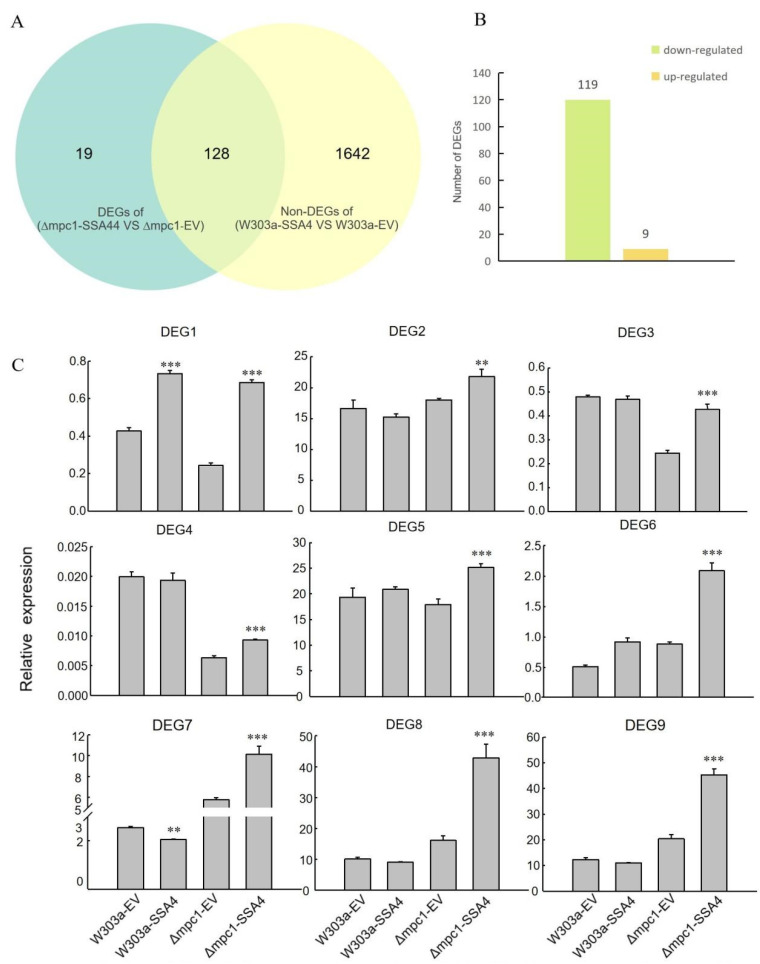
Analysis of the differentially expressed genes (DEGs) identified by RNA-Seq. (**A**) Venn diagram between the DEGs of the ∆mpc1 group (*∆mpc1*-EV VS *∆mpc1*-SSA4) and non-DEGs of the W303a group (W303a-EV VS W303a-SSA4). DEGs were selected with a log_2_ (fold change) > 1 or <−1/2 and padj ≤ 0.05. (**B**) The numer of up- and down-regulated DEGs in the intersection of the Veen diagram. (**C**) Validation of RNA-seq data by qRT-PCR. The comparative 2^−ΔΔCt^ method was used to calculate the relative expression level of the DEGs. The gene number was marked on the top. The paired samples *t*-test methods was used to calculate the significance of the difference. “**” and “***” represent *p* < 0.01and *p* < 0.001, respectively.

**Figure 5 ijms-25-11026-f005:**
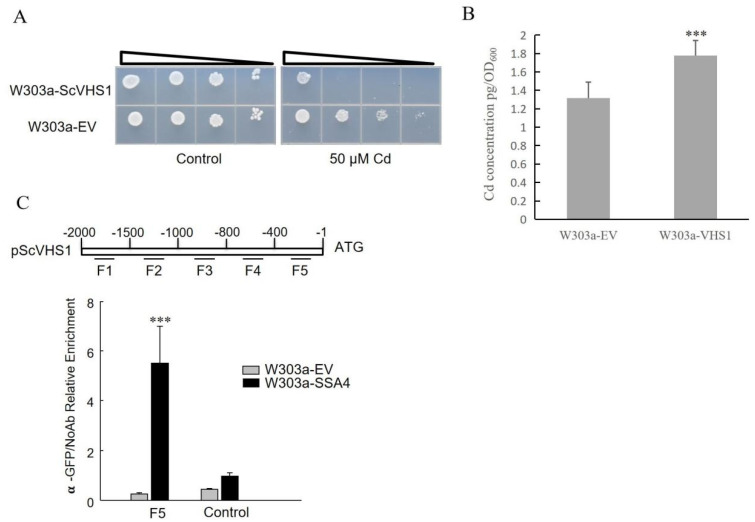
Assessment of Cd tolerance and accumulation in yeast cells overexpressing *VHS1*. (**A**) Dilution bioassay comparing the phenotype of the strain overexpressing *VHS1* and the strain carrying an empty vector under Cd stress. Triangles represent serial 10-fold dilutions, and the starting OD_600_ was 0.3. The experiment was repeated three times with a consistent result. (**B**) Determination of Cd content in yeast cells. The two types of yeast strains were grown on SC solid plates with 50 µM of CdCl_2_ at 30 °C for 5 days. Cells were collected, and the OD_600_ was recorded before the measurements by an atomic absorption spectrometer. (**C**) ChIP-qRCR was carried out to test the interaction between SSA4 and the *VHS1* promoter. DNA immunoprecipitated by a GFP antibody was amplified by qRT-PCR using primers for different regions of the *VHS1* promoter (labeled with F1–F5). Error bars indicate ± SD from three independent experiments. The significance of the difference was calculated using a paired samples *t*-test by SPSS version 20.0 (*** means *p* < 0.001).

**Figure 6 ijms-25-11026-f006:**
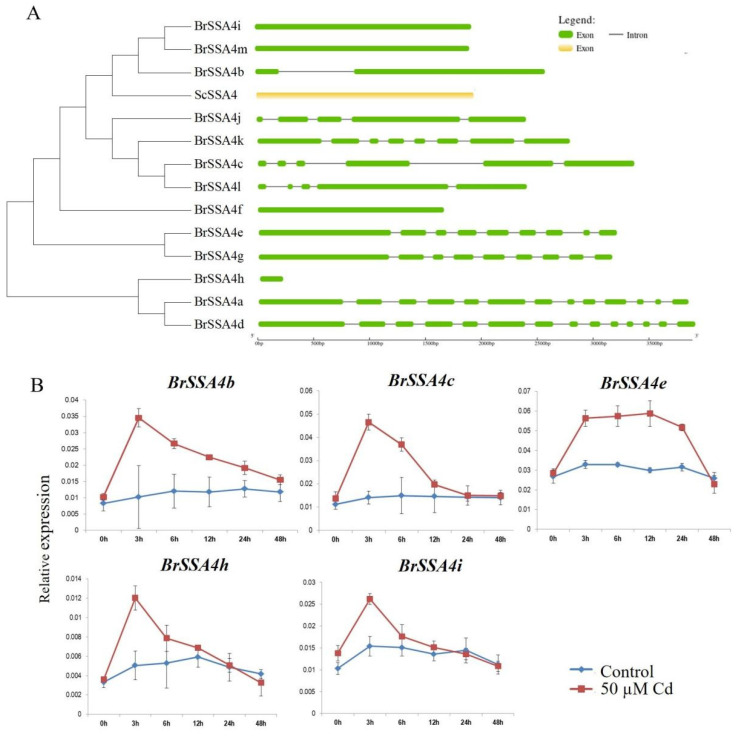
Evolutionary analysis of the SSA4 family members in Chinese cabbage and the expression patterns of the *BrSSA4s* under Cd stress. (**A**) Phylogenetic relationships among the translated BrSSA4 proteins and the intron/exon structure of relative *BrSSA4* genes. The green and yellow sections represent the exons of *BrSSA4s* and *ScSSA4*, respectively, while the black straight line denotes the intron of those genes. (**B**) Expression analysis of the *BrSSA4* genes under Cd treatment. Three-week-old plants were treated with 50 μM CdCl_2_ or same volume distilled water for 0, 3, 6, 12, 24, and 48 h before the leaves were collected. Only members of *BrSSA4s* exhibiting significant differences in expression levels are displayed.

**Figure 7 ijms-25-11026-f007:**
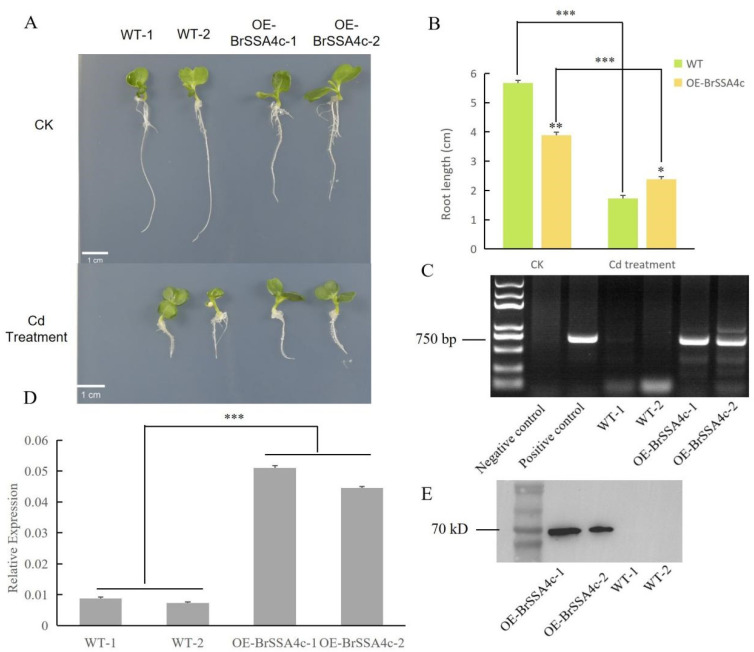
Identification of transgenic Chinese cabbage plants overexpressing *BrSSA4c*. (**A**) Phenotypes of transgenic and wild-type plants grown on half MS solid medium with or without 50 μM CdCl_2_ for 10 days. (**B**) Statistical analysis on root length of seedlings cultured under the growth condition as shown in (**A**). Three seedlings of each class were selected for the measurement of root length. (* *p* < 0.05; ** *p* < 0.01; *** *p* < 0.001). (**C**) Identification of the transgenic seedlings by PCR amplifying vector sequence, in which distilled water and an empty plasmid were used for the negative and positive control, respectively. (**D**) The expression levels of *BrSSA4c* in WT and transgenic seedlings by qRT-PCR. Values were means ± SD (*n* = 3, *** *p* < 0.001). (**E**) Immunoblotting to verify the expression of the GFP-BrSSA4c fusion protein in WT and transgenic Chinese cabbage plants.

**Figure 8 ijms-25-11026-f008:**
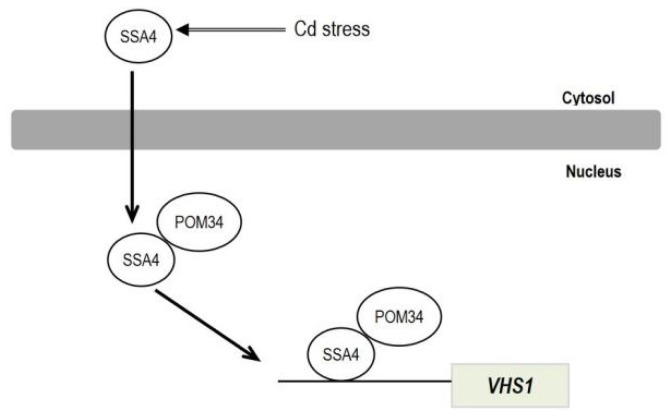
Hypothesis model of ScSSA4 regulation of Cd stress in yeast. Under Cd stress, SSA4 binds with an important component of NPCs named POM34, translocates from the cytoplasm into the nuclear, and regulates the expression of its down-stream gene *ScVHS1*, leading to reduced Cd accumulation in yeast cells.

## Data Availability

All sequence data were deposited in the NCBI Sequence Read Archive (SRA) under accession number PRJNA1157793. Available online: http://www.ncbi.nlm.nih.gov/Traces/sra (accessed on 6 September 2024).

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
