# Peer review of "SSA4 Mediates Cd Tolerance via Activation of the Cis Element of VHS1 in Yeast and Enhances Cd Tolerance in Chinese Cabbage"

_ijms, 2024, doi:10.3390/ijms252011026_

Round 1
Reviewer 1 Report
Comments and Suggestions for Authors
· I suggest to modify the title; Change the “resistance” into “tolerance”
· The abstract is poorly structured. Begin the abstract with a brief background, clearly state the objective of the study, and then summarize the main results. Finally write the main conclusion.
· Please provide information on where the authors obtained the Seeds of Chinese cabbage cultivar “Guangdongzao”. Also cultivar features including cadmium/abiotic stress response
· Please provide details about the soil used, including its composition and fertilization.
· Elaborate on the transcriptome sequencing procedure; information about sequencing platform and filtering steps for raw reads are missing
· qPCR assay: Rewrite this section with clearer information on instruments, replications, and cocktail composition.
· Increase the image size/resolution for figure 2
· Move the table 1 to supplementary information
· Improve the discussion by incorporating the latest references. Rewrite the conclusion with information related how this study significantly advance our knowledge and contribute to improve the Cadmium (Cd) stress tolerance in Cabbage.
· Revise the manuscript to align with the IJMS journal's guidelines by formatting references and other required elements, as it currently does not meet the standards.
· Review the text for English language errors, typos, and grammar before resubmitting.
Comments on the Quality of English LanguageReview the text for English language errors, typos, and grammar before resubmitting.
Reviewer 2 Report
Comments and Suggestions for Authors
The article presents a good approach and successfully explains the interaction of SSA4 as an activator element of Cd-tolerant VHS1. Also, the sequence of experiments used is logical. However, I have found some minor errors that need to be corrected to improve the article.
- Point 2.3., third line, the use of et al. here does not make sense.
- There should be a section 2.14. where the general statistical analysis used (program and version) and the type of statistics for each trial are discussed.
- The figures seem to be well-referenced, and the legends provide enough information to understand them. However, in Figure 3B, it might be worth looking for other images where the interaction between ScSSA4 and ScPOM34 is more visible.
- In Figure 5B, I would check the error bars, as they seem very similar, which could be a mistake.
- Lastly, in Figure 7, I found several errors in the legend.
- In the transcriptomic profile, you could discuss more about the pathways that were mainly affected, as this is only briefly mentioned in the article. While this may not be the most relevant part of the paper, it could open up avenues for future research, both within your group and for others.
- Regarding the discussion, in the sixth paragraph, first line, Brassica juncea should be mentioned, and you could also expand on the evolutionary process of BrSSA4 and consider citing/referencing it in connection with other similar genes or processes.
- I could find some english errors, most of them in the discusion section. Correct it before resend. Some examples: Although kinds various of genes, participate in the absorbance absorption, reduction of the Cd accumulation, which resulting in low Cd accumulation, contorl? control?, can transferred be transferred, there is an interaction between the promoter..., be involved in.
Round 2
Reviewer 1 Report
Comments and Suggestions for Authors
I recommend the manuscript for publication to IJMS